# An Innovative Dual-Axis Precision Level Based on Light Transmission and Refraction for Angle Measurement

**Yubin Huang, Yuchao Fan** **, Zhifeng Lou, Kuang-Chao Fan \*** **and Wei Sun**

School of Mechanical Engineering, Dalian University of Technology, Dalian 116024, China;
huangyubin@mail.dlut.edu.cn (Y.H.); wd_fyc@163.com (Y.F.); Louzf@dlut.edu.cn (Z.L.);
sunwei@dlut.edu.cn (W.S.)

\* Correspondence: fan@ntu.edu.tw

**Abstract:** Currently, the widely used pendulum-type precision level cannot be miniaturized because reducing the size of the pendulum will reduce its displacement so as to decrease the measurement accuracy and resolution. Moreover, the commercial pendulum-type level can only sense one direction. In this paper, an innovative compact and high-accuracy dual-axis precision level is proposed. Based on the optical principle of light refraction and the reference of the invariant liquid level, the pendulum is no more needed. In addition, based on the light transmission design, there is no reflection signal to interfere with the true signal. Therefore, the level can achieve a high accuracy and small-sized design. The calibration result shows the error of the proposed precision level is better than ±0.6 arc-sec in the measurement range of ±100 arc-sec, and better than ±5 arc-sec in the full measurement range of ±800 arc-sec.

**Keywords:** dual-axis level; light refraction; light transmission; angle measurement

---

## 1. Introduction

In precision engineering, with the significant improvement in positioning sensor accuracy in the past few decades [1], the influence of angular error on the accuracy, repeatability, and stability of precision machinery has become increasingly important [2,3]. There are many types of angle measuring instrument that have been applied in precision engineering [4–6]. Some multi-degree-of-freedom measurement systems have been reported in recent years [7–9], and the roll angle measurement of a long stage is still affected by unstable laser beams [10,11]. For the advantage of measuring the roll error of the linear stage, and for the measurement of the inclination angle while installing the precision machinery, it is still necessary to know the precision level.

Traditionally, according to the different measurement principle, spirit levels can be divided into two types: the bubble type and the pendulum type. Due to the disadvantages of measurement resolution and the accuracy of the bubble-type sprit level, the bubble-type sprit level can only meet the requirement of rough adjustment of equipment horizontal installation. Currently, most commercial precision levels are designed based on a pendulum mechanism [4], and usually they are the one-axis pendulum type [4,5]. The pendulum-type precision level is also used as a sensor for monitoring the ground rotation of some large-scale scientific instruments, such as the Laser-Interferometer-Gravitational-wave-Observatory (LIGO) [12,13]. The resolution and accuracy of the mechanical pendulum-type level are dependent on the design of the pendulum and mechanism; for that, the size and weight of the pendulum-type precision level can hardly be reduced [14]. In addition, the design and manufacturing accuracy of the pendulum will directly affect the accuracy of the pendulum-type precision level, and the complex

structure and requirement for high-precision manufacture processing mean that the cost of this type of level cannot be reduced. In order to improve the resolution and accuracy of the pendulum-type precision level, expensive precision sensors such as capacitive sensors further increase the cost of the level. These problems limit the further application of the precision level.

In order to miniaturize the size of the precision level, many novel measurement principles have been proposed. Using gravity as a reference, Ueda [15] proposed a high-precision micro capacitive inclination sensor, and Shimizu [16] proposed a roll error measurement system made by combining two micro capacitive inclination sensors to eliminate the influence of external disturbance. However, the accuracy and resolution of capacitive inclinometers are larger than 1 arc-sec. Based on the grating diffraction and reflection, Gao developed a three-axis autocollimator for detecting three angular error motions of a precision stage [17], and Shimizu [18] modified the system to make a liquid-surface-based three-axis inclination sensor for the measurement of stage tilt motions. However, the system was complex and the accuracy of the measured angles was still larger than 1 arc-sec. The dual-axis pendulum-type precision level was proposed by Fan [19], with the accuracy and resolution both less than 1 arc-sec. A brand-new design of a pendulum-free precision level was proposed by Torng [20] using the light refraction principle and the reference of the invariant liquid level. The dual-axis level could be small in size and light in weight. However, the first reflection beam from the liquid surface would interfere with the refracted beam and produce noise signals. Zhang [21] directly used the surface reflection beam to design a dual-axis level. However, it required high-energy laser power.

In this paper, a modified design of a refraction-type level from the author's previous system [20] is proposed. With the new principle of single-light refraction and transmission, an innovative dual-axis precision level is improved from the previous double refraction type. It can achieve a satisfactory accuracy and resolution in a compact and simple design. The measurement principle will be described in detail in Section 2, and the factors that affected the design of this type of precision level are discussed in Section 3. The calibration and application experiments of the prototype sensor will be presented in Section 4. At the end of this paper, some directions to improve the performance of the proposed precision level are discussed.

## 2. Measurement Principle

The optical configuration of the proposed dual-axis level is shown in Figure 1a. The reference laser emitted from the laser diode (LD) passes through the transparent liquid in a transparent container and is incident on an autocollimator unit composed of a focus lens (FL) and a quadrant photodetector (QPD). The mirrors M1 and M2 are used to reflect the reference laser to reduce the size. When the sensor is inclined, the liquid surface remains level. The reference laser emitted from the transparent liquid into the air is refracted. The angle change of the refracted reference laser is measured by the autocollimator unit. According to Snell's law, the relationship between the inclined angle and the angle change of the reference laser can be expressed as:

$$\begin{aligned} n_1 \sin(\varepsilon_y) &= n_2 \sin(\varepsilon_{ym}) \\ n_1 \sin(\varepsilon_x) &= n_2 \sin(\varepsilon_{xm}) \end{aligned}, \tag{1}$$

where $\varepsilon_y$ is the inclined angle around the Y axis, as shown in Figure 1b, and $\varepsilon_x$ is the inclined angle around the X axis. $\varepsilon_{ym}$ and $\varepsilon_{xm}$ are the refraction angles in the X and Y directions, respectively, and $n_1$ and $n_2$ are the refractive index of the liquid and the air. When the inclined angle is small, the small angle is almost equal to its sine function value, ignoring the very small non-linear error. Equation (1) can be simplified to:

$$\begin{aligned} n_1 \varepsilon_y &= n_2 \varepsilon_{ym} \\ n_1 \varepsilon_x &= n_2 \varepsilon_{xm} \end{aligned}. \tag{2}$$

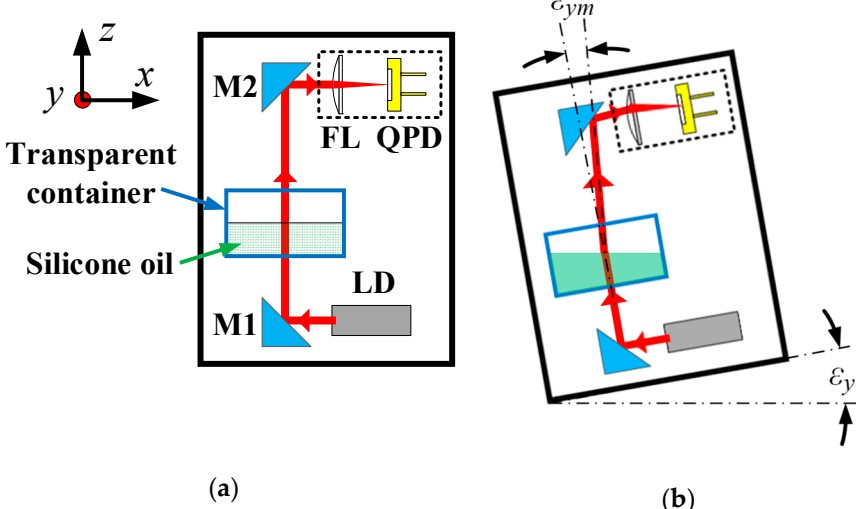

**Figure 1.** Schematic diagram of the dual-axis precision level: (**a**) optical configuration; (**b**) measurement principle.

According to the auto-collimation principle, the change in the reference laser's incident angle from $\varepsilon_{ym}$ to $\varepsilon_{xm}$ will cause the change in light spot's position from $\delta_{yQPD}$ to $\delta_{xQPD}$ on the sensitive surface of the QPD:

$$\varepsilon_{xm} = \frac{\delta_{xQPD}}{f}; \varepsilon_{ym} = \frac{\delta_{yQPD}}{f}, \tag{3}$$

where $f$ is the focal length of FL, the change in the light spot's position from $\delta_{yQPD}$ to $\delta_{xQPD}$ would lead to a change in the light intensity received by each photodiode of the QPD sensor, resulting in a change in the photocurrent output by the QPD. The photocurrents of the QPD are converted to voltages by an I/V conversion circuit. Based on the working principle of QPD and Equations (2) and (3), the relationship between the output voltages of the QPD (V1, V2, V3, V4) and the inclined angle $\varepsilon_y$ and $\varepsilon_x$ can be expressed by:

$$\begin{aligned} \varepsilon_x &= \frac{n_2}{n_1}\varepsilon_{xm} = k_{\varepsilon x}\frac{(V_1+V_4)-(V_2+V_3)}{(V_1+V_2+V_3+V_4)} \\ \varepsilon_y &= \frac{n_2}{n_1}\varepsilon_{ym} = k_{\varepsilon y}\frac{(V_1+V_2)-(V_3+V_4)}{(V_1+V_2+V_3+V_4)} \end{aligned}, \tag{4}$$

where $k_{\varepsilon x}$ and $k_{\varepsilon y}$ are the constants obtained by the calibration process, which will be mentioned in Section 4. The proposed measurement principle is modified from the author's previous work [20] to avoid the influence of the reflected light from the liquid surface on the QPD, which would cause a random error. In addition, compared with the measurement principle using the reflected light from the liquid surface [21], the intensity of the reference laser is lower and the signal of QPD is higher in our proposed measurement principle, leading to the better resolution performance of the proposed system.

It should be noticed that, although the bottom wall of the transparent container will produce the additional refraction of the laser beam, this refraction is always the same regardless of the angle of the level, because the LD, container, and autocollimator are tilted together. The principle of angle measurement by the autocollimator is to detect only the angle change in the incident laser beam, and this angle change is purely from the change in the refraction angle between the liquid surface and the laser beam. In addition, the autocollimator is not sensitive to the beam shift.

## 3. Structure Design

The main structure design of the sensor part of the proposed precision level is shown in Figure 2. The laser diode and micro autocollimator unit are mounted on the lower and upper sides of the container, respectively. A piece of optical window is fixed on the bottom of the container, allowing the reference laser to pass through while preventing the transparent liquid from leaking out. Above the

container, a special design is used to prevent liquid from leaking when the sensor is tilted [20]. The structural design of the sensor is rigid and does not contain any moving parts to increase the stability of the sensor. In order to provide the damping ratio to the dynamic system of the sensor for the stabilization of the measurement signal, the transparent liquid uses silicone oil, which has viscosity and stable physical and chemical properties, which ensures the stable performance of the sensor. The total size of the sensor is determined by the volume of the container, and the design of the container is affected by the following factors.

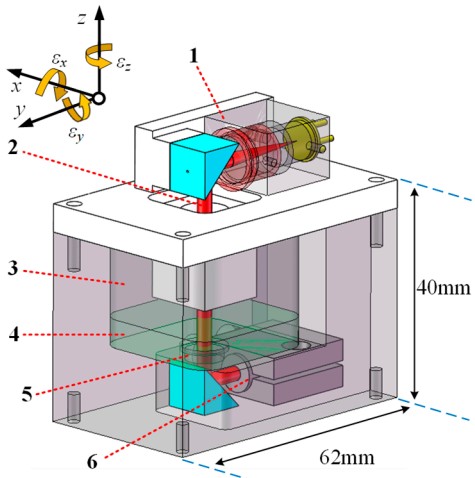

**Figure 2.** Structure design of the refractive precision level sensor: (1) micro autocollimator unit, (2) reference laser beam, (3) container, (4) transparent liquid, (5) optical window, (6) laser diode.

### 3.1. Surface Adhesion Effect

At the solid–liquid interface between the container and the liquid, the liquid surface will bend because of the influence of surface adhesion. As shown in Figure 3, if the refractive interface is concave, the linearity and stability of the sensor signal will be affected. Therefore, the container must have sufficient size and the liquid must have sufficient depth to ensure that the liquid surface which the reference laser passes through is flat. Through a simple analysis, the size of the container's cross-section was designed to be $30 \times 30$ mm$^2$, and the depth of the silicone oil was set to 7 mm.

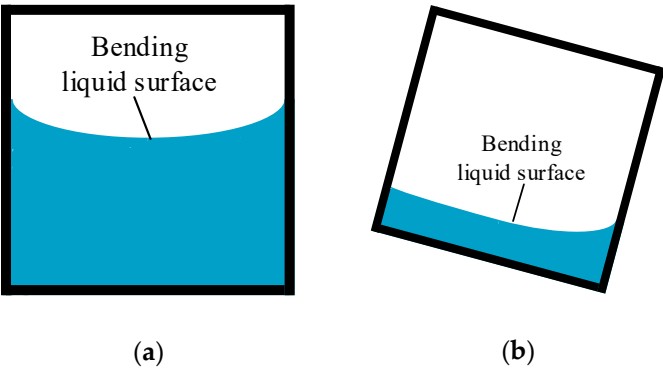

(**a**)　　　　　　　　　　　　　　　(**b**)

**Figure 3.** The influence of the surface adhesion effect. (**a**) Too small container size; (**b**) too shallow liquid depth.

### 3.2. Leakage Prevention Design

When the level is not in use, it will be put in a carrying case. Hence, the container needs to be sealed to prevent liquid leakage. However, if we use the optical window to seal the container on the top, the liquid will inevitably remain on the optical window after the sensor returns from the rest

pose, especially for silicone oils with a higher viscosity. This is the natural phenomenon of surface wettability. That will significantly reduce the measurement accuracy and the stability of the precision level. In order to avoid this problem, the container has been specially designed to ensure that the liquid will not leak out in any posture without using an optical window to completely seal the container [20]. A tube with a certain length is mounted on the top of the container. The hollow tube allows the reference laser to pass through. When the level is tilted, the tube wall can prevent the liquid in the container from leaking out, as shown in Figure 4.

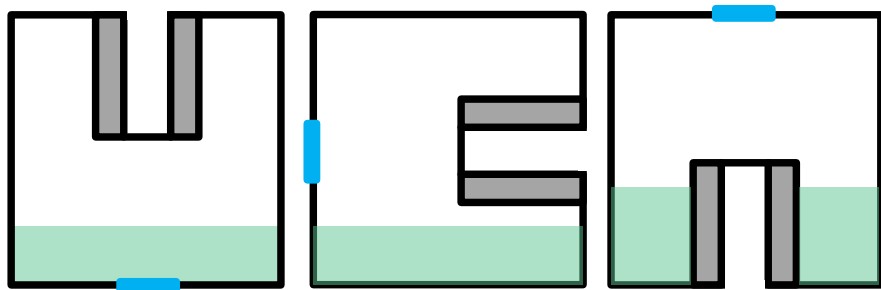

**Figure 4.** The leakage prevention design: liquid in the container in different tilted states.

Considering the influence of the above factors on the structure design, the final designed container size is $30 \times 30 \times 25$ mm$^3$, as shown in Figure 4, and the size of the prototype sensor is $60 \times 37 \times 55$ mm$^3$.

## 4. Calibration and Application Experiments

The prototype of the manufactured dual-axis precision level based on the measurement principle proposed in Section 2 is shown in Figure 5. The level body and the electronic box are separate so that they can be mounted together or separately according to the installation space. The laser diode has a wavelength of 635 nm (model DI635, Huanic, Xian, China,). The micro-autocollimator set, consisting of a high precision QPD (QP5.8-6-TO5, First Sensor, Berlin, Germany) and a focus lens (FL1, φ10, Tokyo, Hitachi, Japan), was constructed to detect 2D angle changes. The signal acquisition electronic device is to acquire and process the signal of the QPD using an analog-to-digital converter (ADC 7606, Analog Devices, Norwood. USA) and a micro-control unit (MCU, ARM SAMD21 Cortex-M0+, Microchip Technology). The measurement data is wirelessly transmitted to the computer via Bluetooth to avoid the influence of pulling the data cable when moving the level.

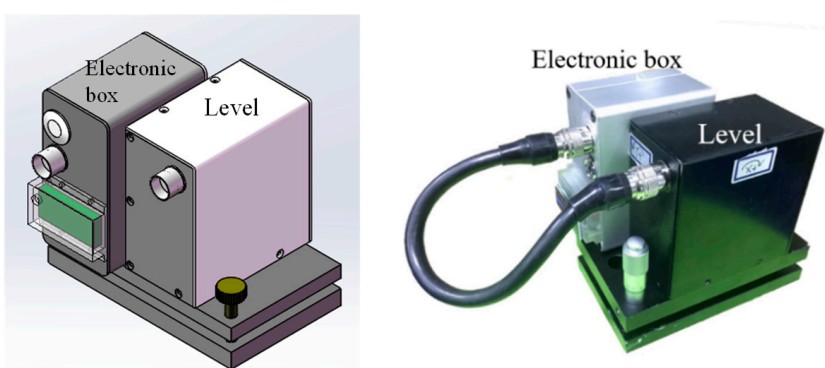

**Figure 5.** The prototype of the dual-axis precision level: 3D design and photo.

*4.1. Calibration Results*

The proposed precision level was calibrated by a commercial autocollimator (AutoMAT Co. model 5000U; resolution: 0.01 arc-sec; accuracy: ±0.1 arc-sec; repeatability: 0.05 arc-sec; uncertainty: 0.119 arc-sec). It is known that the output signal of QPD will show obvious nonlinearity in a large range [22]. In order to obtain the most suitable linear sensitivity coefficient on different measurement

ranges, the two inclined directions of the dual-axis precision level were calibrated for the different measurement ranges of ±100 arc-sec, ±400 arc-sec, and ±800 arc-sec, separately. The results of the averaged linearity curve and residual curve of five times calibration are shown in Figure 6. It can be seen that the residual error increases with the increase in the measurement range, but the peak-to-valley value of residual errors never exceeds 1% of the measurement range. For the measurement range of ±100 arc-sec, the peak-to-valley value of the residual was less than 0.5% of the measurement range. The calibration results show that an accuracy of ±0.6 are-sec in the range of ±100 arc-sec was obtained, and in the full measurement range of ±800 arc-sec, the accuracy was ±4 are-sec. The resolution was 0.1 arc-sec. The overall performance is very satisfying.

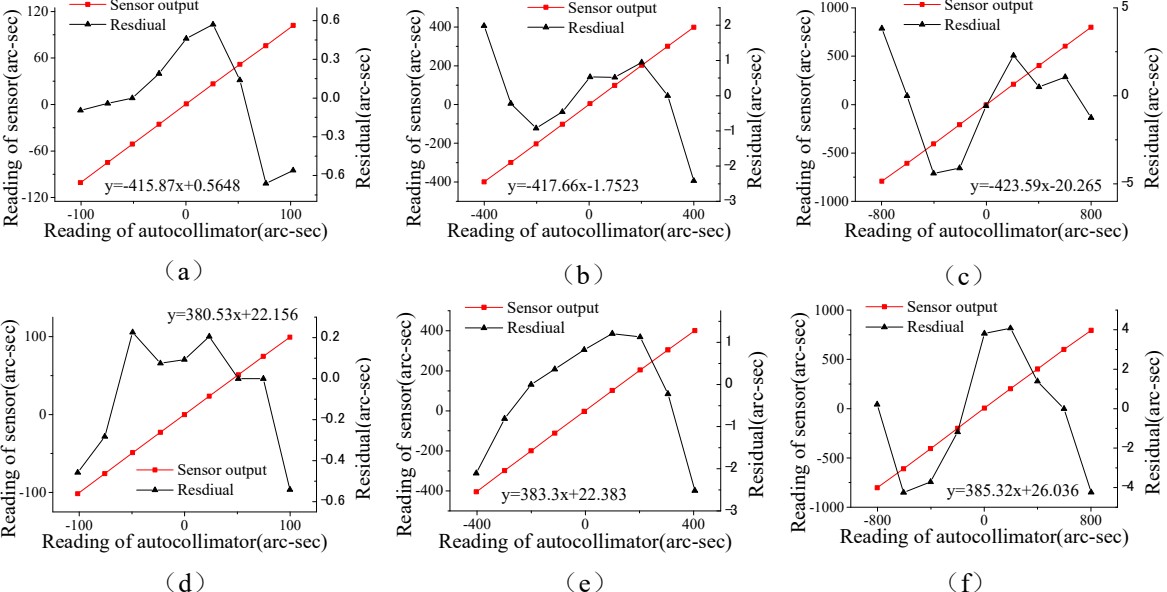

**Figure 6.** Calibration results of the dual-axis precision level: (**a**) $\varepsilon_x$ in ±100 arc-sec; (**b**) $\varepsilon_x$ in ±400 arc-sec; (**c**) $\varepsilon_x$ in ±800 arc-sec; (**d**) $\varepsilon_y$ in ±100 arc-sec; (**e**) $\varepsilon_y$ in ±400 arc-sec; (**f**) $\varepsilon_y$ in ±800 arc-sec.

*4.2. Comparison Measurement Experiment*

To verify the measurement accuracy of the proposed precision level, a comparison measurement experiment was designed. The set-up of the comparison experiment is shown in Figure 7a. The precision level was mounted on a linear stage, and the pitch error of the linear stage was simultaneously measured by the proposed precision level and the autocollimator. After the performance of one direction was tested, the proposed precision level was turned to 90° to compare the other direction. The pitch error of the tested linear stage was adjusted to be close to 100 arc-sec in order to compare the accuracy of the proposed precision level in a large measurement range. The comparison results are shown in Figure 7b. It can be seen that the peak-to-valley value of the residual error is less than 5% of the total pitch error. The experiment results proved that the accuracy of the proposed precision level is acceptable.

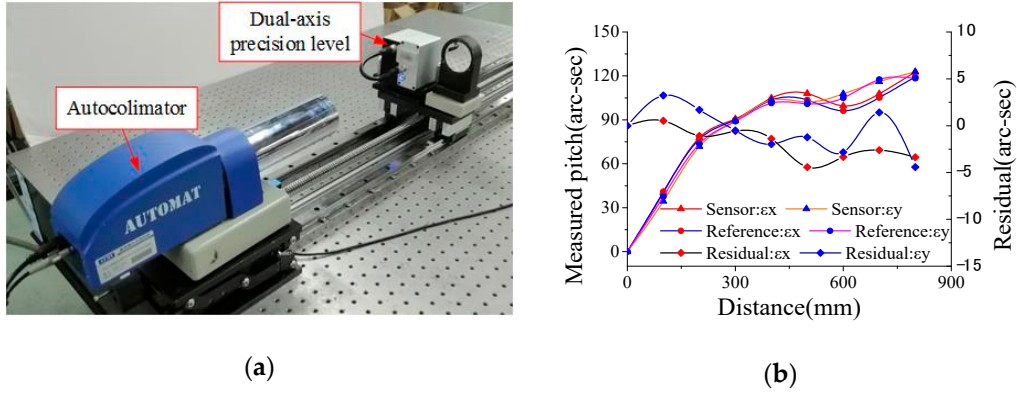

(**a**)                                                                (**b**)

**Figure 7.** Comparison experiment: (**a**) experiment set up; (**b**) results.

### 4.3. Measurement Repeatability Tests

The measurement repeatability of the proposed precision level was obtained by measuring 5 times at 9 positions on a class 00 granite table, as shown in Figure 8. The experiment results are shown in Figure 9. The measurement uncertainty expressed by ±3σ of the dual-axis precision level in the X and Y directions are ±1.1 arc-sec and ±0.92 arc-sec, respectively. This result proves that the measurement repeatability of the proposed precision level is satisfactory.

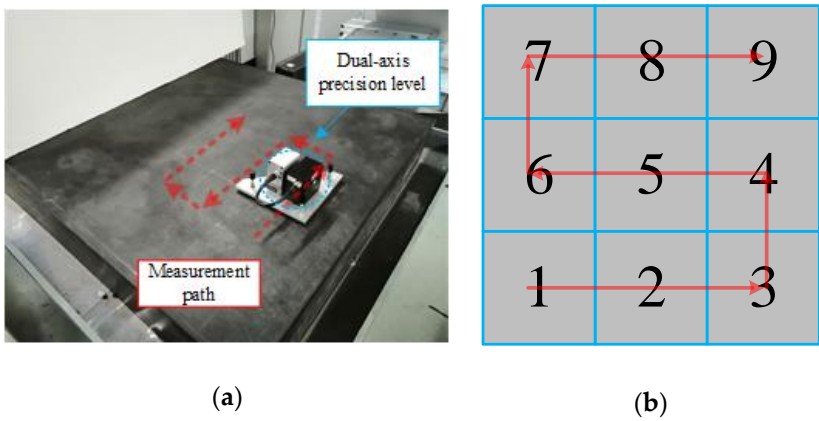

(**a**)                                                                (**b**)

**Figure 8.** Measurement repeatability test: (**a**) photo of the experiment set-up; (**b**) measured positions.

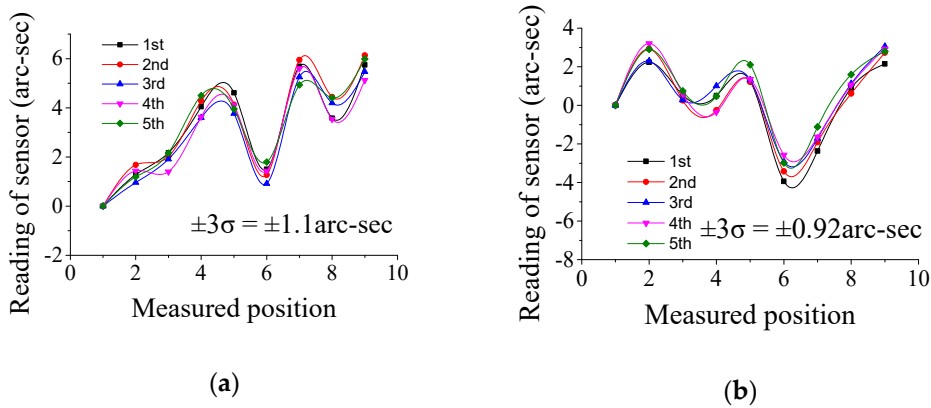

(**a**)                                                                (**b**)

**Figure 9.** Measurement repeatability: (**a**) $\varepsilon_x$; (**b**) $\varepsilon_y$.

### 4.4. Response Time Test

According to the International Standard Organization (ISO) technical report [11], the response time of a precision level is an important performance factor that determines whether the precision level

is practical or not. To verify the response time of the proposed dual-axis precision level, the precision level was fixed on a linear stage of a CNC (computer numerical control) lathe. When the linear stage moved 100 mm at a speed of 60 mm/s, the raw data of the proposed dual-axis precision level from the start to the end of the movement were saved by the software with the sampling rate of 1000 samples per second. When the jitter of the reading of the precision of the dual-axis level was less than 0.1 arc-sec, the reading of the level was considered to be stable. The response data are shown in Figure 10. It can be seen that the settling time to steady-state condition was less than 1 s. This performance is much better than our previous system that needs a 3 s settling time [21].

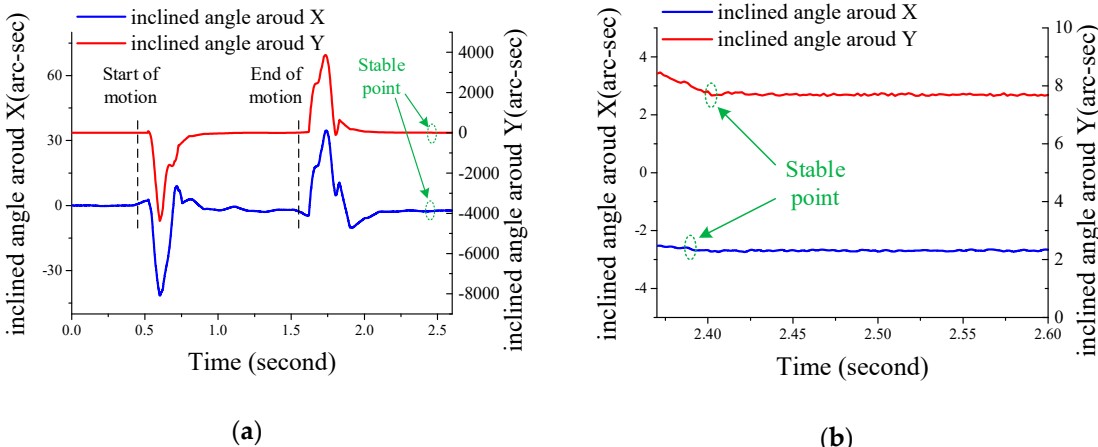

(**a**)                                                                 (**b**)

**Figure 10.** The response time of the proposed precision level: (**a**) all the response raw data; (**b**) partial view.

It should be noted that the response time of the precision level designed according to the measurement principle proposed in Section 2 is affected by the viscosity of the liquid in the container. The filtering algorithm used in the signal processing system would also affect the response time. The liquid used in the prototype of the precision dual-axis level is a synthetic silicone oil with a viscosity of about 200 centistokes. The filtering algorithm used in the signal processing system was a moving average filter, and the length of the sampling window of the moving average filter is 10.

*4.5. Stability Tests*

The stability of the proposed precision level was tested in a non-temperature-controlled environment. While measuring the stability of the precision level, it is directly mounted on an optical bench. The one-hour stability data of the developed level sensor are shown in Figure 11. The drift of the reading is less than 1 arc-sec within one hour. The stability of the proposed dual-axis precision level is satisfactory. It can also be seen from the stability data that the short-term reading jitter is only about 0.1 arc-sec, which can also represent the resolution of the proposed sensor.

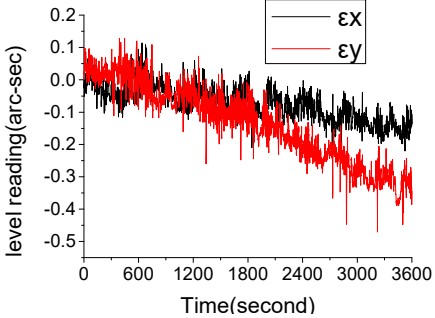

**Figure 11.** Stability data of the proposed precision level.

## 5. Measurement Uncertainty Analysis

Benefiting from the simple measurement principle and the compact structure design of the inclination sensor, the measurement uncertainty of the proposed dual-axis precision level is mainly affected by the performance of the QPD and the electronic device. Thus, it is possible to evaluate the measurement uncertainty of the proposed precision dual-axis level based on the data obtained by the experiment proposed in Section 4. The combined standard measurement uncertainty of the inclination angle around the X axis $u_{\varepsilon x}$ could be obtained by:

$$u_{\varepsilon x} = \sqrt{u_{cal\_x}^2 + u_{res\_x}^2 + u_{drf\_x}^2 + u_{rep\_x}^2} \tag{5}$$

The first influential factor, $u_{cal\_\varepsilon x}$, is the standard uncertainty of the system calibration, which can be obtained from Figure 6. Taking the P-V data of the ±100 arc-sec range and using a rectangular distribution, the standard uncertainty of this part was evaluated to be 0.652 arc-sec. The second factor, $u_{res\_\varepsilon x}$, is the standard uncertainty of the resolution, which can be obtained from the steady-state signal variation in 60 min. From the experiment, the P-V data was 0.09 arc-sec in a rectangular distribution, so that its uncertainty was 0.057 arc-sec. The third factor, $u_{drf\_\varepsilon x}$, is the standard uncertainty of the drift due to environmental effect. According to Figure 11, the maximum drift in the X-direction was 0.22 arc-sec in one hour and was in a rectangular distribution. Its standard uncertainty was 0.127 arc-sec. The fourth factor, $u_{rep\_\varepsilon x}$, is the standard uncertainty of the measurement repeatability, and its probability distribution is Gaussian. From Figure 9a, it was evaluated to be 0.351 arc-sec. Table 1 summarizes the evaluated results of the combined uncertainty from these four sources.

**Table 1.** Uncertainty of the inclination angle around the X-axis of the proposed precision level.

| Sources of Uncertainty | Symbol | Type | Probability Distribution | Standard Uncertainty |
|---|---|---|---|---|
| Calibration of the sensor (including the contribution of systematic error and the uncertainty of the reference autocollimator) | $u_{cal\_\varepsilon x}$ | B | Rectangular | 0.663 arc-sec |
| Resolution of sensor | $u_{res\_\varepsilon x}$ | B | Rectangular | 0.057 arc-sec |
| Drift | $u_{drf\_\varepsilon x}$ | B | Rectangular | 0.127 arc-sec |
| Repeatability | $u_{rep\_\varepsilon x}$ | A | Gaussian | 0.351 arc-sec |
| Standard uncertainty of $\varepsilon_x$ | $u_{\varepsilon x}$ | | | 0.762 arc-sec |

It should be noticed that the uncertainty of calibration $u_{cal\_\varepsilon x}$ has already taken the systematic error sources into consideration, such as the systematic error of the autocollimator measurement method (including the manufacturing error of the focusing lens and the defocus error of the QPD installation), the nonlinearity caused by the initial refraction angle of the reference laser, and the nonlinearity error of the QPD sensor in the large-range error measurement. Therefore, the used optical components must meet a certain manufacturing accuracy, and the standard operating procedure (SOP) should be followed to ensure that the systematic error caused by the assembly errors can be suppressed. The combined standard measurement uncertainty of the inclination angle around the Y axis $u_{\varepsilon y}$ could be obtained in the same way; the evaluated combined standard uncertainty $u_{\varepsilon y}$ and its sources are listed in Table 2.

**Table 2.** Uncertainty of the inclination angle around the Y-axis of the proposed precision level.

| Sources of Uncertainty | Symbol | Type | Probability Distribution | Standard Uncertainty |
|---|---|---|---|---|
| Calibration of the sensor (Including the contribution of systematic error and the uncertainty of reference autocollimator) | $u_{cal\_\varepsilon y}$ | B | Rectangular | 0.454 arc-sec |
| Resolution of sensor | $u_{res\_\varepsilon y}$ | B | Rectangular | 0.057 arc-sec |
| Drift | $u_{drf\_\varepsilon y}$ | B | Rectangular | 0.242 arc-sec |
| Repeatability | $u_{rep\_\varepsilon y}$ | A | Gaussian | 0.301 arc-sec |
| Standard uncertainty of $\varepsilon_y$ | $u_{\varepsilon y}$ | | | 0.599 arc-sec |

As the final result, the combined uncertainties of the proposed precision dual-axis level were evaluated to be 0.762 arc-sec and 0.599 arc-sec in the $\varepsilon_x$ and $\varepsilon_y$ direction, respectively. The expanded uncertainties were thus evaluated to be 1.506 arc-sec and 1.174 arc-sec in the $\varepsilon_x$ and $\varepsilon_y$ direction (coverage factor $k = 2$, 95% confidence), respectively.

## 6. Discussion

Compared with the traditional precision level, the inclination measurement principle based on the light refraction proposed in this report has some significant advantages.

(1) The proposed inclination measurement principle does not require any moving parts, so the systematic error caused by the improper design and manufacturing error of the pendulum or hinge can be avoided.
(2) Compared with the commercial precision level of other inclination angle measurement principle based on the light reflection or refraction of the liquid level surface, the measurement principle proposed in this report if simpler, uses fewer optical components, and effectively avoids the influence of stray light on the measurement accuracy.
(3) The cost of the sensor and light source used is lower, and there are fewer components that require a high manufacturing accuracy; therefore, while the measurement accuracy is ensured, the cost of the proposed precision level can be kept very low, which provides convenience for the application.

On the basis of this research, the final goal is to design a precision level as a sensor to monitor the inclination angle of a precision instrument or machine. To achieve this goal, the following issues still need to be further studied.

(1) A mathematical model of the surface adhesion at the liquid–solid interface should be built and verified. Thus, the design of the sensor can be optimized through the mathematical model, improving the dynamic performance and reducing the size of the inclination sensor.
(2) The range of the inclination angle measurement should be increased by compensating the systematic error of QPD in a large-range measurement or using PSD as the detector in the precision level, under the premise of ensuring the accuracy and resolution of it.
(3) In order to reduce the size and improve the reliability of the proposed precision level, a further integrated optimization design is still necessary.

## 7. Conclusions

An innovative dual-axis precision level is proposed in this paper. Different from the traditional precision level based on the pendulum measurement principle, the proposed dual-axis precision level is based on the light refractive principle. Compared with the traditional measurement principle of the precision level, this novel inclination angle measurement principle can achieve a high resolution and accuracy at a lower cost and compact design. The proposed light refractive and transmission measurement principle can significantly reduce the size of the precision level while maintaining the measurement precision and accuracy. A prototype precision level sensor was made to verify the feasibility of the instrument. The results of the verification experiments are satisfactory. The resolution of the proposed precision level is 0.1 arc-sec, and the measurement uncertainty is less than ±1.2 arc-sec. The calibration results show that, for the measurement range of ±100 arc-sec, the accuracy of the proposed level is within ±0.6 arc-sec; for the full range of ±800 arc-sec, the accuracy of the proposed level is within ±4 arc-sec. Benefitting from the simplicity of the measurement principle, the accuracy and uncertainty of the proposed precision level are almost determined by the photoelectric sensor and its signal processing electronic device, making it easy to improve its performance. In addition, with a proper size design and response test, the quick settling time of about 1 s exhibits an advantage in practical applications. Future works will concern the angular error measurement of precision machines. At the same time, rigorous research on the surface adhesion effect will help us understand how this

phenomenon affects the error source and dynamic characteristics of this type of inclination angle sensor. The establishment of a mathematical model could help to optimize the design of the proposed precision level, which will also be the focus of future research.

**Author Contributions:** Conceptualization, K.-C.F. and Y.H.; methodology, K.-C.F. and Y.H.; software, Y.F; validation, Y.H., Y.F.; resources, Z.L.; data curation, Y.H; writing—original draft preparation, Y.H.; writing—review and editing, K.-C.F.; supervision, W.S.; funding acquisition, K.-C.F. and Z.L. All authors have read and agreed to the published version of the manuscript.

**Funding:** This research was funded by the fund of The National Key Research and Development Program of China (2017YFF0204800) and the Liaoning Provincial Fund (No. 2020JH6/10500017).

**Conflicts of Interest:** The authors declare no conflict of interest.

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
