# Peer review of "An Innovative Dual-Axis Precision Level Based on Light Transmission and Refraction for Angle Measurement"

_applsci, doi:10.3390/app10176019_

Round 1

Reviewer 1 Report

The experimental principles, evaluation methods and consideration for the results are not valid. There is insufficient information on the experimental methods and experimental conditions. Thus, this manuscript is needed to be rewritten. Other questions and comments are shown in the below.

  1. I have a question about the principle of the proposed sensor. In the manuscript, the optical path variation of the reference beam was modeled as one time refraction. According to Figs. 1 and 2, the reference beam is refracted at least three times in the transparent container and silicone oil before it is incident on the QPD. Therefore, it is assumed that the incident angle of the reference at each interface is different. However, in that manuscript, the position shift of the reference beam due to the tilting of the transparent container is indicated as the angular change of beam above the liquid level, i.e., a single refraction. I couldn't understand the validity of that approximation. The purpose of the proposed sensor is precision level with sub-arc second uncertainty. It is necessary to explain the reference beam shift caused by the refraction of air-transparent container interface and the transparent container-silicone oil interface, respectively. If not, the principle of the proposed method is not valid.

  1. I cannot understand the purpose of the experiment shown in Fig. 6. Why was the number of measurement points set to the same even though the measurement range was expanded? The resolution of the autocollimator is not being effectively used. Figs. 6(b) and (c) include the range of Fig. 6(a), however, the number of measurement points is scarce and comparison is not possible. How many times have the experimental results shown in Figure 6 been repeated? Since there is no error bars on each plot, it is impossible to distinguish between nonlinearity and variation.

  1. How much are the refractive index of the transparent container and silicone oil?

  1. As for the experiment shown in Fig. 10, is it possible to show the speed chart of the linear stage of the CNC lathe? In my opinion, since the water surface of silicone oil is affected by acceleration of the linear stage, agreement between the velocity change and the sensor output is essential for validation.

  1. What kind of device was used to record the sensor output for all measurements? In addition, I would like to know the resolution and sampling frequency of those experiments.

  1. Please correct the misspelling in Fig 1. “Transpartent” is “transparent”. “Silicon oil” is “Silicone oil”.

Reviewer 2 Report

The paper presents a novel and interesting design of a precise dual-axis level. Although the concept is neat and results are promising, there are some important aspects which should be better explained or covered in order to get this paper accepted. These mostly concentrate on a proper handling of uncertainty. This is a well covered topic in any measurement application and should be compliant with a standard like JCGM 100:2008 (GUM) or any other well-established and recognized method in the industrial and/or scientific community.

Here are some key issues which should be addressed in the revised version:

  1. Abstract: generally language side is not great. Repetitions should be avoided like "Moreover".  
  2. Introduction: Measurement of level is an important aspect of metrology. It should be, however, better introduced to the readers. What are other techniques used to measure level, starting from spirit levels through autocollimators and interferometers to pendulum-based solutions. The motivation of this paper and importance of this work should better explained. This section is very limited and should be significantly expanded.
  3. Measurement principle: Simplifying relations (1) to (2) leads to some non-linearity errors. This should be better discussed in the paper.
  4. Measurement principle: Quadrant photodetector should be introduced and explanations of equation 4 should be given.
  5. Measurement principle: what is the effect of the transparent container on additional diffraction of laser beam? Its optical properties are close to air/vaccum?
  6. Structure design: Effect of surface tension has to be enumerated, modelled, analyzed and discussed. This would make this paper more scientific than what it is now - a technical note.
  7. Calibration and application experiments: There is no uncertainty measurement analysis! This is critical for any new metrological design. Callibration procedure should be explained in more details, following well-established standardized methods.
  8. Paper lacks discussion and conclusions are explicitly derived from results. That is not what makes good scientific paper. It requires more context, well-developed discussion, critical analysis and referencing to other existing works.
  9.  

Reviewer 3 Report

Some suggestions are provided to the authors to improve the quality of this work.

Main issues:

  1. Title: Please consider including the keyword: “angle measurement”
  2. Introduction section should be improved. Please consider including a comparative table about other solutions and measurement principles. Also, possible applications.
  3. Please add a section about the optoelectronic circuit used to obtain the output voltages: type of photodiode, laser wavelength, signal conditioning, microcontroller, etc. An electronic circuit of the electronic box will be welcome.
  4. Stability in the lab is fine, however for precision machines could be worse. If possible, discuss this issue with a real application.

Minor issues:

  1. Figure 6: “Resdiual” > Residual
  2. How was the residual error obtained?
  3. In figure 2, please add the dimensions of the structure.
  4. Line 113: “leakage” > Leakage

Round 2

Reviewer 1 Report

The authors provide clear answers to the questions. Furthermore, the addition of section 5 is extremely useful for understanding the validity of the measurement. I still have several questions in the revised manuscript, so I show the questions and comments in the below.

  1. In section 5, can the authors indicate a mathematical model of the measurement error? Eq. (5) indicates the combined uncertainty, but, it is better to show a mathematical model of the measurement error to explain the validity of the choice of standard uncertainty source.

  1. In lines 95-100 of the revised manuscript, I could understand why the authors thought it was single reflection. However, it is presumed to be valid when the beam is incident on a flat surface at an incident angle of 0°. Thus, are the flatness of the bottom surface of the transparent container and the alignment accuracy of the beam incident angle an uncertain source in the measurement? In Section 5, an assembly accuracy of the level and alignment errors of the optical components are not considered as uncertainty sources. There is no explanation that their effects can be removed by the calibration procedure.

  1. Is reflected light generated at the solid-liquid interface or gas-liquid interface? Also does it affect the measurement error?

Reviewer 2 Report

Just some other issues:

  1. Still no introduction is given of how level can be measured. One general paragraph would be strongly appreciated.
  2. The uncertainty was evaluated based on autocollimator measurements which own uncertainty should be considered as well.
  3. Discussion is still weak and superficial.
